# ResearchGPT: Benchmarking and Training LLMs for End-to-End Computer Science Research Workflows

## Abstract

As large language models (LLMs) advance, the ultimate vision for their role in science is emerging: we could build an AI collaborator to effectively assist human beings throughout the entire scientific research process. We refer to this envisioned system as *ResearchGPT*. Given that scientific research progresses through multiple interdependent phases, achieving this vision requires rigorous benchmarks that evaluate the *end-to-end* workflow rather than isolated sub-tasks. To this end, we contribute **CS-54k**, a high-quality corpus of scientific Q&A pairs in computer science, built from 14k CC-licensed papers. It is constructed through a scalable, paper-grounded pipeline that combines retrieval-augmented generation (RAG) with multi-stage quality control to ensure factual grounding. From this unified corpus, we derive two complementary subsets: **CS-4k**, a carefully curated benchmark for evaluating AI's ability to assist scientific research, and **CS-50k**, a large-scale training dataset. Extensive experiments demonstrate that CS-4k stratifies state-of-the-art LLMs into distinct capability tiers. Open models trained on CS-50k with supervised training and reinforcement learning demonstrate substantial improvements. Even 7B-scale models, when properly trained, outperform many larger proprietary systems, such as `GPT-4.1`, `GPT-4o`, and `Gemini 2.5 Pro`. This indicates that making AI models better research assistants relies more on domain-aligned training with high-quality data than on pretraining scale or general benchmark performance. We release CS-4k and CS-50k in the hope of fostering AI systems as reliable collaborators in CS research.

## 1 Introduction

With the rapid development of large language models (LLMs), their potential as research assistants has become increasingly evident (Luo et al., 2025; Zhang et al., 2025b; Eger et al., 2025). Currently, these systems mainly help with literature search or code generation, but their potential is not limited to this. Instead, they could be developed to become general-purpose collaborators capable of assisting the entire scientific workflow, from problem framing to method development and empirical analysis. We refer to this paradigm as *ResearchGPT*. Realizing this vision relies on the availability of *robust benchmarks and training corpora* that can faithfully measure and continuously improve models' ability to assist scientific research.

However, existing benchmarks for LLMs' scientific capabilities remain fragmented, addressing only isolated stages of the research workflow. Generally speaking, prior efforts can be divided into two categories: evaluation-as-exam and evaluation-as-agent. The first line extends the exam-style paradigm, framing evaluation as scientific question answering. For example, SuperGPQA (Du et al., 2025) and Humanity's Last Exam (HLE) (Phan et al., 2025) provide expert- and graduate-level assessments that combine multi-domain and multi-skill challenges approximating human competence. The second line treats LLMs as autonomous research agents, emphasizing workflow automation and task execution. Representative efforts include CSR-Bench (Xiao et al., 2025), DataSciBench (Zhang et al., 2025a), and DSBench (Jing et al., 2024), which focus on data analysis and code deployment tasks. As research is inherently multi-phase and interdependent, a research benchmark should evaluate the end-to-end workflow. Existing efforts fall short of this goal.

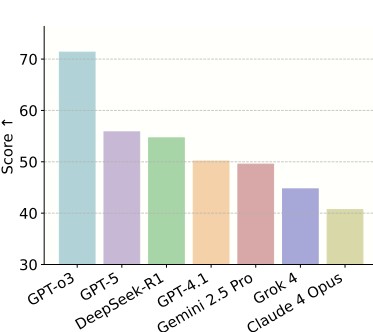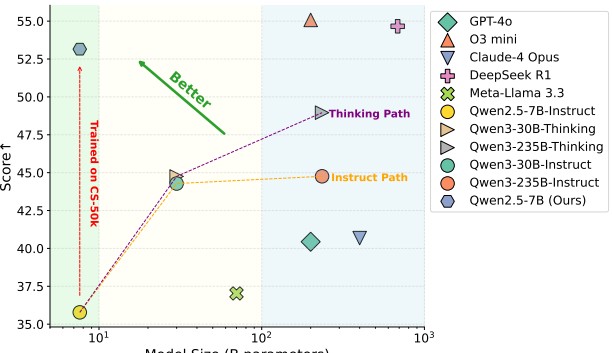

Figure 1: **Benchmarking Comparison on CS-4k.** **Left:** Overall scores of selected models, illustrating clear performance differences. **Right:** A comparison of model size vs. score, where some proprietary model sizes are estimates (Sakthi, 2025; Mashouf, 2025; Abacha et al., 2024). The red arrow highlights the substantial gain from training Qwen on CS-50k. The orange and purple dashed lines indicate the scaling trends of instruction-tuned and reasoning-oriented models.

To address this gap, we introduce **CS-54k**, a high-quality corpus of scientific Q&A pairs in computer science, together with a *reproducible, paper-grounded pipeline* to continuously build and extend such datasets. This pipeline harvests 14k CC-licensed papers from top-tier CS conferences and integrates RAG with multi-stage quality control to ensure that all Q&A pairs are grounded in an authentic scientific context rather than synthetic templates. Research questions are inherently exploratory and should be answered in precise scientific language, requiring models to demonstrate both scientific reasoning and clear expression. Therefore, we formulate the task as *open-ended scientific question answering*, spanning multiple facets of the research workflow and grounded in the paper context, enabling end-to-end evaluation of AI research assistants.

From this unified corpus, we derive two complementary subsets through careful sampling and curation. One is **CS-4k**, a benchmark for rigorous evaluation. It enables holistic and realistic assessment of models' ability to assist the full research workflow, as illustrated in Figure 1. The other is **CS-50k**, a large-scale training dataset addressing the scarcity of domain-aligned data faithfully reflecting scientific workflows. Existing efforts (Liu et al., 2025) highlight LLMs' potential but suffer from insufficient domain-specific corpora, and CS-50k provides extensive and reliable supervision for systematic fine-tuning on authentic research tasks. For models already trained for research, the whole 54k corpus can also be used as a comprehensive evaluation.

Using CS-50k, we fine-tune `Qwen2.5-7B-Instruct` (Yang et al., 2024; Team, 2024) with supervised fine-tuning (SFT) and reinforcement learning via GRPO(Shao et al., 2024). Experiments demonstrate substantial improvements over strong baselines, with dual-reward optimization further mitigating reward hacking and enabling a 7B-scale model to match or surpass many proprietary reasoning-oriented systems, such as `GPT-4.1`, `GPT-4o`, and `Gemini 2.5 Pro`. This suggests that progress in building effective AI research assistants is driven more by domain-specific, high-quality training than by pretraining scale (corpus size/model size) or general benchmark performance. Our main contributions are summarized as follows:

- We introduce **CS-4k**, the first benchmark that systematically evaluates the end-to-end research workflow in computer science through *open-ended scientific question answering*, offering a rigorous yardstick to assess LLMs' ability to assist scientific research.

- We develop a reproducible *paper-grounded dataset pipeline* that integrates RAG and multi-stage quality control for authentic scientific grounding. It is scalable to larger corpora and extendable to future multimodal scientific benchmarks.

- We demonstrate that, in the context of scientific workflows, domain-aligned high-quality training could be more important than pretraining scale or general benchmark performance.

## 2 RELATED WORK

### 2.1 BENCHMARKS FOR SCIENTIFIC AI

With the rapid development of large language models (LLMs), a wide range of benchmarks have been proposed to evaluate their capabilities across different domains and tasks. Early benchmarks in NLP were often built around relatively simple and domain-specific tasks, such as corpus annotation, sentiment classification, relation extraction, or fact-based question answering (Kim et al., 2003; Li et al., 2016; Luan et al., 2018; Jurgens et al., 2018; Jin et al., 2019). As LLMs progressed, benchmark design gradually shifted toward broader and more challenging evaluations, covering both general-purpose reasoning and discipline-oriented examinations. General-purpose evaluations such as BIG-Bench (Srivastava et al., 2023) and HELM (Liang et al.) provide broad coverage across domains, while exam-style datasets including MMLU (Hendrycks et al.), GPQA (Rein et al., 2024), SuperGPQA (Du et al., 2025), and Humanity's Last Exam (HLE) (Phan et al., 2025) test knowledge and reasoning at different academic levels. Scientific reasoning benchmarks such as CURIE (Cui et al., 2025) and SFE (Zhou et al., 2025) further emphasize multi-step problem solving and multimodal understanding in scientific contexts. Recent initiatives like ResearchBench (Liu et al., 2025) and MOOSE-Chem2 (Yang et al., 2025) examine hypothesis generation and fine-grained scientific reasoning, while systems such as Agent Laboratory (Schmidgall et al., 2025) and the AI Scientist (Lu et al., 2024) push toward autonomous end-to-end research workflows.

### 2.2 LLM ALIGNMENT

LLM alignment aims to ensure that model outputs align with human intentions.(Bai et al., 2022; Rafailov et al., 2023; Dong et al., 2023). Early attempts relied on supervised fine-tuning (SFT) on limited, human-annotated data, which improved instruction-following but was constrained by the coverage and quality of the data (Wei et al.; Wang et al., 2023). InstructGPT (Ouyang et al., 2022) then introduced the now-standard RLHF paradigm—combining SFT with a reward model and PPO—to achieve stronger alignment. More recently, Generalized Reward Policy Optimization (GRPO) (Shao et al., 2024) has emerged as a more efficient alternative, streamlining the RLHF pipeline by eliminating the value model and stabilizing optimization. A persistent challenge in this line of work is *reward hacking* (Gao et al., 2023a), where models exploit imperfections in reward models, leading to degraded true alignment. Mitigation strategies such as ensemble rewards and constrained objectives have been proposed (Coste et al.; Moskovitz et al.). While prior alignment efforts emphasized dialogue and safety, here we apply SFT and GRPO on our proposed **CS-50k** to align models with end-to-end scientific research workflows.

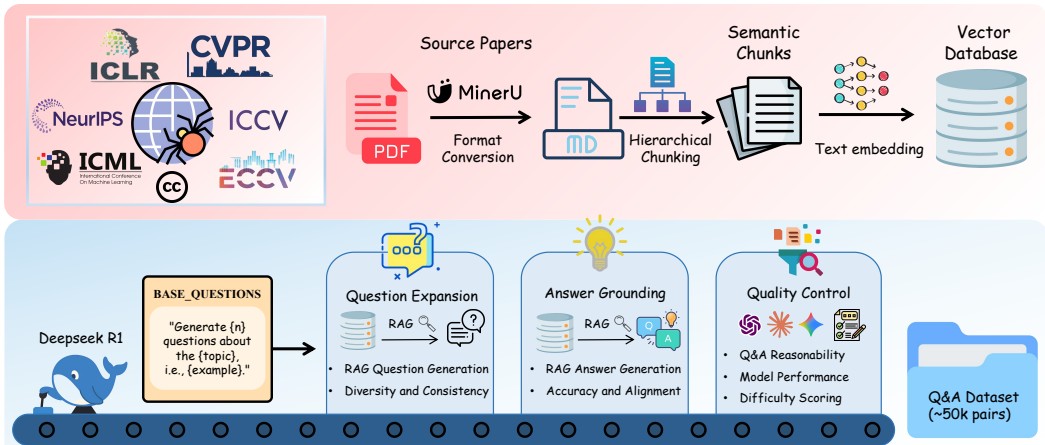

Figure 2: **Dataset construction pipeline.** **Top:** source papers from six CS conferences are converted, chunked, and embedded. **Bottom:** question expansion, answer grounding, and multi-stage quality control yield approximately 50k high-quality Q&A pairs.

## 3 DATASET

To construct a high-quality dataset for evaluating and training LLMs in scientific research tasks, we design a systematic pipeline that spans from large-scale paper collection to fine-grained question–answer (Q&A) generation and quality filtering. The overall process is illustrated in Figure 2.

### 3.1 DATA COLLECTION

We begin by collecting approximately 66,000 papers from arXiv, restricted to six premier computer science conferences: ICML, ICLR, NeurIPS, ICCV, ECCV, and CVPR. Among these, only papers released under Creative Commons open-access licenses are retained, resulting in a final corpus of 14,474 papers and ensuring both legal compliance and reproducibility.

Each paper is converted from PDF format into Markdown using the MinerU (Wang et al., 2024; He et al., 2024) toolkit. This conversion facilitates text-level operations such as segmentation and semantic analysis. To preserve contextual integrity, we apply hierarchical chunking, dividing each paper into semantically coherent segments.

We embed each chunk using the nomic-ai/nomic-embed-text-v1.5 (Nussbaum et al., 2024) model and store the representations in a vector database. This enables efficient retrieval of relevant passages during question generation and ensures that the dataset construction process remains grounded in the original paper content. For each paper, we categorize the content into a set of eight topics that capture the essential components of the scientific workflow, as summarized in Table 1.

| Class | Explanation |
|---|---|
| Research domain | The field or area of study the research addresses |
| Previous methods | Approaches or algorithms previously proposed in related work |
| Existing challenges | Limitations, gaps, or open problems identified in prior research |
| Motivation | The rationale or justification for conducting the research |
| Findings/Assumptions | Key observations or assumptions that guide the research |
| Methods | Proposed approaches or frameworks designed to solve the identified problems |
| Experimental settings | Details of the experimental design, setup, data preparation, or parameter configurations |
| Experimental results | Outcomes and performance reported from experiments or evaluations |

Table 1: Research categories used for topic annotation and their explanations.

### 3.2 DATA GENERATION

Following the segmentation and annotation of papers into topic categories, we proceed to generate question-answer (Q&A) pairs. Instead of generating questions heuristically, we adopt a *retrieval-augmented generation (RAG)* (Lewis et al., 2020; Gao et al., 2023b; Zhao et al., 2024; Fan et al., 2024) pipeline. This approach allows us to anchor both the questions and answers directly to the original paper content, thereby minimizing hallucination and ensuring factual grounding.

For each topic class, we design a *base question template* to guide the question generation process. The template is: "Generate {n} questions about the {topic}, i.e., {example}." This template provides controlled guidance on the type of questions expected within each category, ensuring consistency and relevance across the dataset. Here, {topic} corresponds to the eight research categories in Table 1, and {example} is instantiated using their explanations.

Based on these templates, question construction is conducted in two stages:

1. **Question Expansion.** First, we apply the QUESTION_AUG_PROMPT (Appendix A.1) to expand the template into several diverse drafts. Then, each draft is grounded into a fully specified question through a RAG process, using the QUESTION_GEN_PROMPT (Appendix A.1) together with relevant chunks from the vector database. This step ensures a

variety of questions while maintaining alignment with the original paper content, and RAG guarantees that all generated questions remain relevant and contextually grounded.

2. **Answer Grounding.** Each candidate question is paired with an answer generated through the ANSWER_GEN_PROMPT(Appendix A.1), where retrieved chunks of the paper serve as contextual evidence. This guarantees that the answers are directly supported by the original text, minimizing the risk of hallucination and ensuring factual correctness. RAG plays a key role in ensuring that the generated answers are grounded in the paper's content, aligning both questions and answers in a cohesive manner.

Through this two-stage RAG-driven process, we obtain approximately **600,000 preliminary Q&A pairs**, covering a broad range of aspects from background knowledge to methodological details and experimental findings.

### 3.3 QUALITY CONTROL

To ensure the reliability and usability of the dataset, we design a three-stage quality control pipeline that aligns with our illustrative framework:

- **Q&A Reasonability.** We employ a model-based evaluation (DeepSeek-R1) to automatically assess Q&A pairs and filter out unreasonable or low-quality samples. The evaluation relies on a dedicated prompt (Appendix A.2) that guides the model in checking semantic coherence and factual consistency. This process reduces the dataset to around **100k high-quality Q&A pairs**, which are then standardized and reformatted for consistency.

- **Model Performance.** To further refine the dataset, we conduct evaluations using multiple strong LLMs, including GPT-4-mini, google/gemini-2.5-flash-preview, and claude-3-5-haiku-latest. For each Q&A pair, we compute the model response score of these models' responses, using a prompt (Appendix A.2) that explicitly queries model correctness. Questions that are consistently answered correctly by all models (too trivial) or consistently answered incorrectly (too difficult or ambiguous) are removed.

- **Difficulty Scoring.** To characterize question complexity, we apply the LLMDifficulty ScoreFilter(Appendix A.2) module from the Data-Juicer (Chen et al., 2024a;b) library. This step assigns difficulty scores to each question and is used to ensure that the train/test splits maintain a balanced distribution across difficulty levels.

### 3.4 DATA STATISTICS

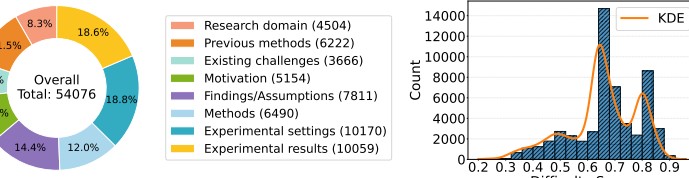 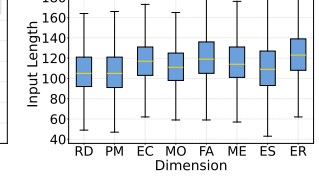

(a) Category distribution  (b) Difficulty distribution  (c) Input length distribution

Figure 3: Overall dataset statistics for CS-4k/50k combined. (a) Category coverage across 8 research workflow dimensions. (b) the distribution of difficulty scores estimated by an LLM-based scorer (histogram with KDE fit). (c) Input length variation across categories.

To facilitate a robust evaluation of the models, we split the dataset into two parts: **CS-50k** (training set) and **CS-4k** (test set). Approximately 4k samples are selected from the entire dataset for CS-4k, with the remainder forming CS-50k. The split preserves the distribution of research categories, difficulty levels, and input lengths, ensuring balanced coverage in both training and test sets, while maintaining access to the full paper corpus. This section provides an overview of key statistics for the constructed dataset, including its scale, category distribution, and difficulty composition.

Table 2 reports the distribution of the initially collected 66k papers across six premier computer science conferences, covering machine learning, computer vision, and artificial intelligence. After filtering for Creative Commons open-access licenses, 14,474 papers are retained for dataset construction.

The overall dataset consists of approximately **54k high-quality Q&A pairs** spanning eight research workflow categories: *research domain, previous methods, existing challenges, motivation, findings/assumptions, methods, experimental settings, and experimental results* (Figure 3a). Each question is assigned a continuous difficulty score estimated by an LLM-based assessor, yielding a smooth spectrum from straightforward factual queries to challenging reasoning tasks (Figure 3b). Input length distributions show moderate variation across categories without extreme skew (Figure 3c). Overall, the dataset is large-scale, diverse, and balanced, providing a reliable benchmark for evaluating LLMs on scientific research workflows. Additional statistics for the train (CS-50k) and test (CS-4k) splits are given in Appendix B.

Table 2: Statistics of dataset sources.

| Conference | Count |
| --- | --- |
| NeurIPS | 20,286 |
| ICML | 10,979 |
| ICLR | 11,679 |
| CVPR | 11,842 |
| ICCV | 5,369 |
| ECCV | 6,166 |

## 4 EXPERIMENTS AND EVALUATIONS

In this section, we present experiments to evaluate the effectiveness of the proposed **CS-4k** dataset and the performance of state-of-the-art large language models (LLMs) trained using our dataset. Specifically, we aim to assess: (1)The ability of LLMs to handle end-to-end research workflows across diverse scientific tasks. (2)The performance of fine-tuned models (SFT and GRPO) trained on **CS-50k** compared to baselines evaluated on **CS-4k**.

### 4.1 EXPERIMENTAL SETUP

We conduct a comprehensive evaluation of the state-of-the-art LLMs on CS-4k. For reasoning models, we evaluate: DeepSeek-R1(Guo et al., 2025), GPT-5(OpenAI, 2025b), o3, o4-mini(OpenAI, 2025c), o3-mini(OpenAI, 2025d), o1-mini(OpenAI, 2024c), Gemini 2.5 Pro, Gemini 2.5 Flash(Team et al., 2023), Claude 4 Opus, Claude 4 Sonnet, Claude 3.7 Sonnet(Anthropic, 2023), Grok4, Grok3 mini fast, Grok3 mini(xAI, 2023), Qwen3-30B-A3B-Thinking-2507, Qwen3-235B-A22B-Thinking-2507(Team, 2025). For chat models, we evaluate: GPT-4.1, GPT-4.1-mini(OpenAI, 2025a), GPT-4o(OpenAI, 2024b), GPT-4o-mini(OpenAI, 2024a), Gemini 2.0 Flash(Team et al., 2023), Claude 3.5 Haiku(Anthropic, 2023), Grok3(xAI, 2023), Llama-3.3-70B-InstructAI@Meta, 2024, Qwen3-30B-A3B-Instruct-2507, Qwen3-235B-A22B-Instruct-2507(Team, 2025)

**Training settings.** We select **Qwen2.5-7B-Instruct** as the backbone model and train it on the proposed **CS-50k** dataset. All experiments are conducted on 8×NVIDIA A100-80G GPUs. Detailed hyperparameter configurations are provided in Appendix C.

For supervised fine-tuning (SFT), we employ the **LLaMA-Factory** framework (Zheng et al., 2024), training for three epochs and selecting the checkpoint with the lowest validation loss. For reinforcement optimization (GRPO), we adopt the **verl** framework (Sheng et al., 2024), training for three epochs on top of the SFT checkpoint. Based on this setup, we explore the following strategies:

- **SFT only.** Supervised fine-tuning on CS-50k to align outputs with ground-truth answers.
- **SFT + GRPO (single-RM).** GRPO optimization on top of SFT using **Qwen2.5-7B-Instruct** as the single reward model (RM).
- **SFT + GRPO (dual-RM).** To mitigate reward hacking, GRPO employs two independent reward models, **Qwen2.5-7B-Instruct** and **Llama-3.1-8B-Instruct**, whose scalar rewards are averaged to form the final training signal.

**Metrics.** When benchmarking, we configure all models' temperatures to 0 for reduced randomness and employ a unified zero-shot prompt template across all tasks. To assess answer quality, we adopt an **LLM-as-a-Judge** paradigm, using `gpt-4.1-2025-04-14` as the evaluation model.

The scoring prompt, provided in Appendix A.2, instructs the judge to assign an integer score from 0 to 10 based on semantic and technical alignment with the reference answer. Each model prediction is assessed along eight predefined research-oriented categories: **Research domain**, **Previous methods**, **Existing challenges**, **Motivation**, **Findings/Assumptions**, **Methods**, **Experimental settings**, and **Experimental results**. In addition, we report an **Overall** score that averages across all categories to provide a comprehensive view of model performance.

## 4.2 RESULTS AND ANALYSIS

We evaluate the performance of diverse LLMs on the proposed **CS-4k** benchmark using the LLM-as-a-Judge score as the evaluation metric. The results in Table 3 demonstrate that CS-4k effectively reveals fine-grained differences in model capabilities.

| Model | Research domain | Previous methods | Existing challenges | Motivation | Findings/ Assumptions | Methods | Experimental settings | Experimental results | Overall |
|---|---|---|---|---|---|---|---|---|---|
| *Reasoning Models* | | | | | | | | | |
| DeepSeek-R1 | 55.93 | 53.80 | 57.08 | 61.09 | 57.16 | 55.91 | 48.04 | 54.42 | 54.66 |
| GPT-5 | 59.08 | 58.54 | 66.75 | 65.05 | 60.79 | 63.19 | 40.10 | 51.39 | 55.83 |
| o3 | **71.66** | **71.05** | **78.83** | **77.56** | **75.71** | **74.40** | **59.57** | **72.07** | **71.35** |
| o4-mini | 65.04 | 64.36 | 69.34 | 72.38 | 68.84 | 67.61 | 54.25 | 65.68 | 64.90 |
| o3-mini | 56.26 | 53.95 | 54.49 | 59.95 | 57.61 | 56.73 | 48.58 | 56.50 | 55.08 |
| o1-mini | 42.88 | 40.71 | 41.79 | 46.94 | 44.96 | 44.32 | 34.33 | 43.54 | 41.93 |
| Gemini 2.5 Pro | 49.58 | 48.28 | 48.28 | 56.76 | 53.32 | 51.56 | 40.46 | 52.00 | 49.54 |
| Gemini 2.5 Flash | 45.67 | 42.68 | 45.40 | 51.74 | 47.16 | 47.43 | 36.36 | 43.58 | 44.17 |
| Claude 4 Opus | 45.61 | 39.70 | 43.43 | 50.21 | 44.21 | 45.51 | 30.09 | 38.06 | 40.68 |
| Claude 4 Sonnet | 43.86 | 40.04 | 44.01 | 49.04 | 43.42 | 44.84 | 31.21 | 37.86 | 40.48 |
| Claude 3.7 Sonnet | 41.63 | 36.46 | 39.89 | 48.08 | 41.79 | 41.81 | 29.55 | 33.50 | 37.79 |
| Grok 4 | 45.99 | 44.79 | 47.08 | 50.91 | 47.26 | 47.06 | 36.46 | 45.03 | 44.73 |
| Grok 3 mini fast | 44.45 | 42.25 | 43.76 | 49.72 | 45.16 | 45.78 | 34.93 | 42.45 | 42.75 |
| Grok 3 mini | 44.78 | 41.31 | 43.21 | 49.66 | 44.56 | 44.90 | 38.02 | 42.15 | 42.96 |
| Qwen3-30B-A3B-Thinking-2507 | 43.32 | 41.91 | 52.77 | 57.36 | 46.68 | 45.12 | 34.53 | 46.38 | 44.75 |
| Qwen3-235B-A22B-Thinking-2507 | 51.63 | 47.27 | 59.45 | 61.84 | 51.64 | 52.14 | 35.14 | 48.21 | 48.96 |
| *Chat Models* | | | | | | | | | |
| GPT-4.1 | 51.25 | **49.01** | 49.56 | 54.15 | 53.33 | 52.74 | 42.11 | 52.52 | 50.15 |
| GPT-4.1-mini | 49.73 | 47.55 | 47.08 | 53.11 | 51.40 | 51.48 | 42.20 | 50.78 | 48.85 |
| GPT-4o | 42.37 | 39.72 | 38.32 | 45.39 | 42.77 | 42.26 | 34.30 | 41.47 | 40.44 |
| GPT-4o-mini | 40.95 | 38.28 | 37.45 | 43.42 | 40.63 | 40.78 | 32.97 | 41.25 | 39.13 |
| Gemini 2.0 Flash | 40.33 | 37.79 | 39.67 | 44.15 | 40.50 | 42.18 | 28.03 | 34.96 | 37.29 |
| Claude-3.5 | 28.81 | 25.19 | 30.77 | 23.70 | 20.53 | 28.05 | 16.77 | 16.10 | 22.12 |
| Llama-3.3-70B-Instruct | 39.41 | 36.39 | 35.36 | 41.84 | 39.20 | 39.09 | 30.75 | 37.86 | 37.03 |
| Qwen3-30B-A3B-Instruct-2507 | 44.90 | 43.09 | 44.85 | 50.03 | 47.49 | 45.88 | 36.72 | 45.70 | 44.28 |
| Qwen3-235B-A22B-Instruct-2507 | 46.50 | 43.80 | 45.88 | 53.21 | 46.94 | 46.44 | 37.99 | 43.93 | 44.76 |
| **Qwen2.5-7B-Instruct** | 38.87 | 35.71 | 35.55 | 41.66 | 37.71 | 38.56 | 30.21 | 33.86 | 35.78 |
| **Qwen2.5-7B-Instruct-sft** | 42.55 | 40.26 | 43.83 | 47.98 | 47.45 | 44.09 | 35.22 | 46.31 | 43.11 |
| **Qwen2.5-7B-Instruct-GRPO** | 48.96 | 45.88 | 46.53 | 56.92 | 57.06 | 51.19 | **43.46** | 56.31 | 50.97 |
| **Qwen2.5-7B-Instruct-GRPO-2RMs** | **51.75** | 48.26 | **50.36** | **61.97** | **59.09** | **55.31** | 42.94 | **57.63** | **53.15** |

Table 3: Dimension-wise evaluation results of all models on CS-4k, covering eight aspects of the research workflow and the overall score. Higher values indicate better performance.

**CS-4k cleanly separates capability tiers of SOTA LLMs in scientific research.** On CS-4k, the strongest recent models form a clear top tier—for example, o3(71.35) and o4-mini(64.90)—well above mid-tier such as GPT-5 (55.83) and DeepSeek-R1 (54.66), while general chat variants cluster lower (e.g., GPT-4.1 50.15; GPT-4o 40.44; Gemini 2.0 Flash 37.29). This separation is consistent across dimensions and highlights that the benchmark can resolve fine-grained capability gaps.

Beyond absolute tiers, CS-4k also reveals systematic differences within model families. A common pattern is that *reasoning-oriented* variants consistently outperform their *instruction-tuned* counterparts—for instance, Qwen3 "Thinking" models surpass the corresponding "Instruct" versions at both 30B and 235B scales. Figure 1 highlights two distinct scaling trajectories within the Qwen family, contrasting reasoning-oriented and instruction-oriented variants.

(i) **Instruct Path** (orange dashed line) shows diminishing returns, with larger Qwen3-Instruct models plateauing at nearly the same performance level, suggesting that pure instruction-following quickly saturates.

(ii) **Thinking Path** (purple dashed line), in contrast, continues to benefit from scale: even medium-sized Qwen3-Thinking models already rival larger Instruct counterparts, and further scaling widens this gap.

Taken together, these results indicate that reasoning-focused training not only yields stronger models at fixed scales but also scales more effectively, a trend that holds robustly in scientific research tasks.

**CS-4k reveals clear performance differences across research dimensions.** Although the strongest models reach high overall scores, their performance is far from uniform across dimensions of the research workflow. For example, o3 achieves the highest results on most dimensions (78.83 on *Existing Challenges* and 77.56 on *Motivation*), indicating its strength in capturing conceptual framing and problem formulation. However, its score on *Experimental Settings* drops to 59.57, reflecting consistent difficulties in reproducing fine-grained procedural details such as dataset splits, hyperparameter schedules, and hardware specifications. This disparity is systematic: models perform better on high-level reasoning (*Existing Challenges*, *Motivation*, *Methods*) but decline on technical recall (*Experimental Settings*, *Experimental Results*).

These findings suggest that current LLMs are more adept at summarizing and reasoning about high-level research narratives than at faithfully reconstructing the technical underpinnings of scientific experiments. This has important implications for their use as research assistants: while they can provide valuable insights in framing and conceptual reasoning, they remain unreliable for tasks demanding precise reproduction of experimental protocols. We further analyze typical failure cases of this type in our case study (Section 4.2).

**CS-50k training with SFT and GRPO significantly improves model research knowledge.** Our experiments on `Qwen2.5-7B-Instruct` highlight the effectiveness of the CS-50k training corpus and the complementary roles of supervised and reinforcement learning. The base model achieves an overall score of 35.78 on CS-4k, underscoring the limitations of open-source chat-style systems when evaluated on research-oriented tasks. Fine-tuning with supervised learning (SFT) on CS-50k lifts the score substantially to 43.11 (+7.33), as the model learns to align outputs with high-quality, paper-grounded answers. This demonstrates that even relatively small models can acquire significant research knowledge when trained on data faithfully reflecting the scientific workflow.

Building on this foundation, reinforcement optimization with GRPO further improves performance to 50.97 (+7.86 over SFT). Reinforcement signals provide preference guidance that goes beyond factual correctness, emphasizing methodological clarity and experimental rigor. At the category level, the gains are especially notable in *Motivation* and *Findings/Assumptions*, where SFT already enhances recall of research details but GRPO deepens reasoning and coherence, showing how preference optimization complements supervised learning.

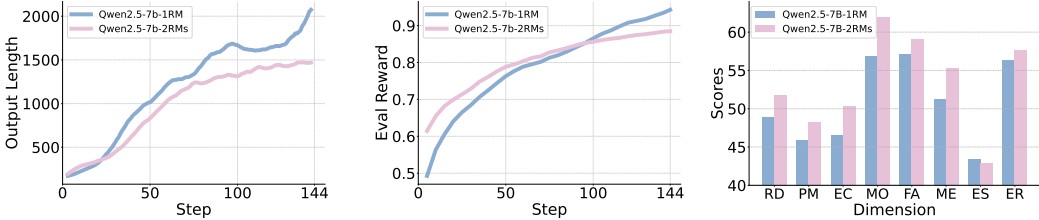

Figure 4: Training and evaluation statistics with single vs. dual reward models: output length curves, evaluation reward curves, and CS-4k scores across research dimensions.

**Mitigating reward hacking.** To mitigate the challenge of reward hacking, we introduce a dual reward model configuration, combining Qwen2.5-7B-Instruct and Llama-3.1-8B-Instruct as parallel evaluators. This setting reduces bias from any single evaluator and encourages more generalizable optimization. With this design, the model reaches 53.15 overall, establishing the strongest performance among all chat models. Despite its modest 7B scale, the system not only closes the gap with but even surpasses most proprietary reasoning-oriented counterparts, highlighting the practical effectiveness of CS-50k training combined with robust reinforcement optimization.

Figure 4 further illustrates the mechanism behind this improvement. While single-reward (1RM) optimization yields higher in-training reward, it also drives a sharp increase in output length (left), a hallmark of reward hacking. By contrast, dual-reward (2RM) training maintains lower but more stable rewards(middle) and effectively controls output length. Crucially, these differences translate

into better generalization at evaluation time: when tested on CS-4k with *GPT-4.1 as an independent reward model*, the 2RM system consistently outperforms the 1RM counterpart (right), with the largest gains in *Motivation* and *Findings/Assumptions*. This confirms that higher in-training reward does not necessarily imply stronger alignment, and that dual-reward optimization provides a more faithful and robust alignment signal.

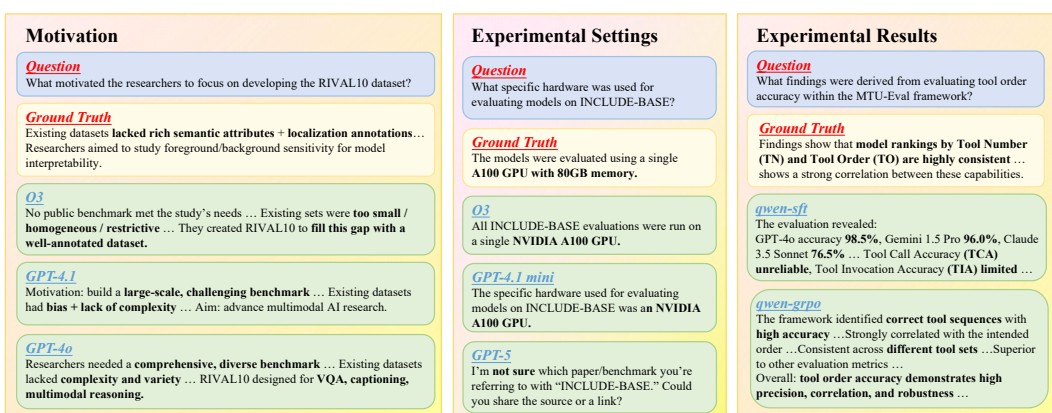

Figure 5: Case studies on CS-4k across different research workflow dimensions.

**Case study.** To better understand the behaviors revealed by CS-4k, we present illustrative examples across three representative dimensions.

*Motivation.* On the rationale for RIVAL10, o3 cited missing semantic attributes and localization annotations. GPT-4.1 noted dataset complexity, while GPT-4o gave only a vague claim of a "comprehensive benchmark." This indicates that reasoning-oriented models capture research motivations more faithfully, while chat models fall back on generic narratives.

*Experimental Settings.* For questions on hardware configurations in the INCLUDE-BASE benchmark, o3 and GPT-4.1-mini both identified the use of a single NVIDIA A100 (80GB). GPT-5, however, failed to answer and instead asked for clarification. This highlights a common weakness: many models struggle to ground responses in technical setups essential for reproducibility.

*Experimental Results.* For result-comparison questions, SFT-trained `Qwen` gives concise summaries but often hallucinates—for example, fabricating numbers or misattributing baselines in the MTU-Eval case. GRPO variants still make minor numerical errors but capture key findings such as tool correlations and consistency across settings. This reveals a trade-off: SFT favors brevity but risks distortion, while GRPO produces longer outputs that better preserve factual and logical alignment.

## 5 CONCLUSION AND DISCUSSION

We introduced **CS-4k**, the first benchmark for evaluating models' ability to assist end-to-end computer science research workflows, and **CS-50k**, a companion training dataset. Both are built through a *reproducible, paper-grounded pipeline* that harvests 14k CC-licensed papers and integrates RAG with multi-stage quality control, resulting in over 50k high-quality Q&A pairs. Experiments show that CS-4k stratifies capability tiers of state-of-the-art models, while training on CS-50k with SFT and GRPO substantially enhances open models. Together, they provide both a rigorous yardstick and a practical resource for the advancing scientific AI.

Looking ahead, the same pipeline naturally extends to multimodal scientific evaluation. Beyond text, future work will incorporate figures and tables, enabling grounded Q&A generation through VLM-RAG and supporting the assessment of *VLM agents* across tasks and modalities. These extensions will allow models to reason over heterogeneous scientific evidence and handle more realistic research scenarios. We envision this direction as a step toward building reliable multimodal AI collaborators that can assist, accelerate, and eventually transform the process of scientific discovery.

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

## LLM USAGE

Large language models (LLMs) were used solely to polish the writing (e.g., grammar correction and phrasing improvements). They did not contribute to research ideation.

## A    DATA CONSTRUCTION

Our dataset is constructed through two types of prompts: **data generation prompts** and **quality control prompts**. Each prompt is designed to ensure and diversity in the generated data.

### A.1    DATA GENERATION PROMPTS

We design three types of prompts to guide data generation.

---

**❶ Question Augmentation Prompt**

You are a skilled linguistic assistant specializing in query transformation. When presented with a user's query, your objective is to meticulously rephrase and enhance it by:

- Utilizing synonyms, alternative phrasings, and varied sentence structures
- Adjusting grammatical form (e.g., converting commands to questions) while preserving intent
- Restructuring information flow without losing original components
- Improving clarity and natural fluency
- Maintaining all factual elements and nuanced meaning

Avoid adding new information or omitting key aspects. Prioritize conversational tone and logical coherence in your revisions.

**Format Requirements:**

- Provide only the rewritten query
- No supplementary explanations or formatting

**Example Transformation:**

    **Original:** Give me 5 reasons of sleeping early

    **Response:** Can you list five advantages of maintaining an early bedtime schedule?

**Now transform this query while adhering to all guidelines:**

    **Original:** {input}

    **Response:**

---

**❶ Question Generation Prompt**

**Context information is below.**

--------------------

{context}

--------------------

**Objective**: Provide self-contained questions using **ONLY** the given context. Formulate responses as if explaining to someone **UNAWARE** of this context in a JSON list. When addressing domain-specific concepts:

- Explicitly state the domain upfront
- Avoid all references to "context" or "provided information"
- Assume zero prior knowledge

---

- Avoid vague or generalized references like "this research" or "your research area." Instead, directly address the specific subject matter or topic being discussed.
- For example, instead of asking "What is the motivation of this work?", you might consider framing the question more specifically, such as "What motivated the researchers to focus on developing [specific subject]?" or "Why [specific subject] matters?"

**Response Constraints**:

- **FORBIDDEN** generalized phrases: ["provided context", "given context", "above information", "mentioned context", ...]; replace them with specific subject matter
- For ambiguous terms (e.g., "xxx"):
    - First establish domain context
    - Then provide explanation
- Responses are required to be formatted as a JSON-parseable list: `["response1", "response2", ...]`
- Maintain consistent JSON syntax without explanatory text
- Absolute prohibition of:
    - JSON/text hybrids
    - Context-existence references
- Ensure that the generated responses contain only a JSON list of several plain text questions without including codes, markdowns, or anything else.

**Example Response (MUST be formatted as a JSON-parseable list):**
`["Question1", "Question2", "Question3", "Question4", "Question5"]`

**Query:** {input}
**Answer:**

---

## ℹ **Answer Generation Prompt**

**Context information is below.**

--------------------

{context}

--------------------

Using **STRICTLY** the provided context text above, follow these prioritized rules to answer the query:

**[PRIORITY 1: CONTENT PRESERVATION]**

1. **DIRECT QUOTATION PRINCIPLE:**

   - First attempt: Use **VERBATIM SENTENCES** from context that directly answer the query
   - When multiple relevant sentences exist:
      a. Preserve original sequence unless illogical
      b. Link using minimal transitional phrases (e.g., "Furthermore", "This shows")
      c. Replace context-specific references (e.g., "this study", "our method") with `[PROPOSAL]`

**[PRIORITY 2: CONTEXTUAL SYNTHESIS]**

2. If no single sentence fully answers but context contains relevant information:

   a. Combine MULTIPLE context fragments using:
      - Only coordinating conjunctions (and / but / or)
      - Basic punctuation (commas, semicolons)
   b. PRESERVE original wording from source material

    c. Remove non-essential modifiers (e.g., "interestingly", "as shown in Table 1")

    d. Maintain factual integrity without interpretation

**[STRICT PROHIBITIONS]**

- NEVER:
  - Introduce information beyond context boundaries
  - Add explanations not explicitly stated
  - Paraphrase in ways that alter original meaning
  - Speculation beyond context
  - Include personal interpretations

**[FALLBACK INSTRUCTIONS]**

- If context contains partial but insufficient information: State existing relevant facts **PRECISELY** using original wording
- Only output `"The context does not contain relevant information"` when:
  - Zero contextual connection exists
  - All potential answers require speculation

**[OUTPUT REQUIREMENTS]**

- Maximum preservation of original lexical choices
- Grammatical coherence for standalone understanding

**Query:** {input}
**Answer:**

## A.2 QUALITY CONTROL PROMPTS

To evaluate the quality of generated data, we use several prompts for correctness checking, difficulty assessment, and answer scoring.

---

**⊙ LLM Answer Evaluation Prompt**

You will be given a question and an answer. Judge whether the answer is reasonable for the question.

If the answer is relevant and correctly or appropriately answers the question, output 1. If the answer is irrelevant, nonsensical, incorrect, or does not address the question, output 0. Only output 0 or 1, and nothing else.
Now, judge the following:

    **Question:** {question}

    **Answer:** {answer}

    **Output:**

---

**⊙ LLM Difficulty Score Prompt**

You are an expert pedagogical evaluator for LLM training data. Analyze each data sample through multiple difficulty lenses and provide calibrated scores with detailed reasoning. Follow these guidelines:

**1. Evaluation Dimensions** Rate each dimension (1–5 scale: 1=Novice-friendly, 3=Intermediate, 5=Expert-level):

- Linguistic Complexity: Vocabulary sophistication & syntactic structures

- Conceptual Depth: Abstraction level & theoretical requirements

- Prior Knowledge: Required domain-specific understanding

- Step Complexity: Problem-solving steps needed

- Ambiguity: Multiple valid interpretations

**2. Output Format**

```json
{
  "dimension_scores": {
    "linguistic_complexity": ,
    "conceptual_depth": ,
    "prior_knowledge": ,
    "step_complexity": ,
    "ambiguity":
  },
  "flags": ["multistep_reasoning", "cultural_context", ...],
  "rationale": "Technical analysis of challenge sources"
}
```

**3. Special Instructions**

- Differentiate intrinsic vs. extrinsic difficulty factors

- Account for varying cultural/educational backgrounds

- Mark samples requiring cross-domain knowledge synthesis

- Consider temporal aspects for time-sensitive subjects

- Flag ambiguous samples needing difficulty bracketing

- Response a JSON dict

**Example Response:**

```json
{
  "dimension_scores": {
    "linguistic_complexity": 3,
    "conceptual_depth": 5,
    "prior_knowledge": 4,
    "step_complexity": 4,
    "ambiguity": 5
  },
  "flags": ["nonlinear_reasoning", "semantic_ambiguity"],
  "rationale": "High conceptual difficulty due to multi-layered
  metaphor interpretation requiring philosophy background.Moderate
  linguistic complexity offset by implicit cultural references."
}
```

⊙ **Prompt for evaluation**

You are a grading expert. Judge whether the final answers given by the candidates below are consistent with the reference answers, i.e., whether the candidates answered correctly. Provide a single **integer score from 0 to 10**.

**Scoring Guide:**

9–10: **Fully correct, detailed, and clearly structured.** The prediction matches the reference not only in meaning, but also in technical detail, terminology, and step-by-step structure. Only assign this score if the answer is nearly indistinguishable from the reference.

7–8: Covers all key points and includes all key technical details, but may have slight simplifications, structural looseness, or phrasing issues. The prediction is still technically sound and semantically aligned.

5–6: Covers nearly all major technical points with acceptable clarity and structure. The prediction may slightly oversimplify or omit minor supporting details, but the core reasoning is intact.

3–4: Incomplete or partially correct. Misses one or more core concepts or steps, or is expressed in a vague, confusing, or disorganized way.

1–2: **Weak or flawed answer.** Contains major factual errors, misunderstands the question, or mixes unrelated ideas. May reference a few relevant terms, but lacks meaningful explanation or structure.

0: **Fundamentally incorrect or completely off-topic.** No meaningful alignment with the reference content, or content is nonsensical or missing.

**Output Format:**

- Respond with a **single integer from 0 to 10**.
- Do not include any explanation or additional text.

**Special Instructions:**

- The model prediction may contain the reasoning process; you should spot the final answer from it.
- Do not re-answer the question yourself.
- Assign a high score only if the prediction matches the answer **semantically and technically**, considering variations in format.
- Deduct points for missing key technical details or excessive generalizations. Even if the tone is correct, factual or structural omissions should lead to a reduced score.
- Be strict with general or vague answers: If the prediction only provides a high-level overview but omits key technical details, steps, or quantitative findings present in the reference, score **no higher than 6**.
- Ignore minor differences in formatting, capitalization, or spacing.

**Example Response:** 7

**Now start your task.**

> **Question:** {question}
>
> **Reference Answer:** {answer}
>
> **Model Prediction:** {prediction}

# B ADDITIONAL DATASET STATISTICS

We report category, difficulty, and input length distributions for CS-50k (train) and CS-4k (test), showing comparable coverage with the test set slightly skewed toward harder examples (Figures 6–8).

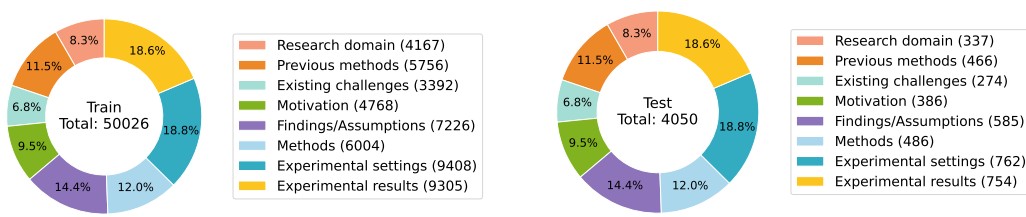

(a) Category distribution (CS-50k / train)          (b) Category distribution (CS-4k / test)

Figure 6: Category distributions for the training and test splits.

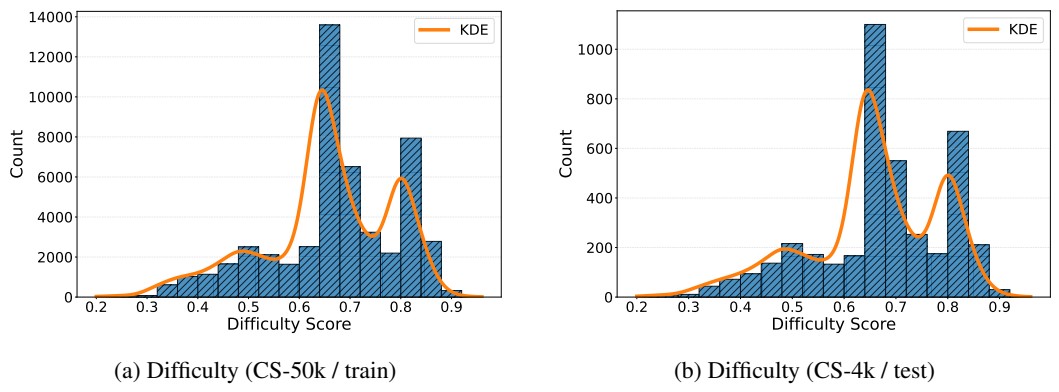

(a) Difficulty (CS-50k / train)          (b) Difficulty (CS-4k / test)

Figure 7: Difficulty distributions for the training and test splits.

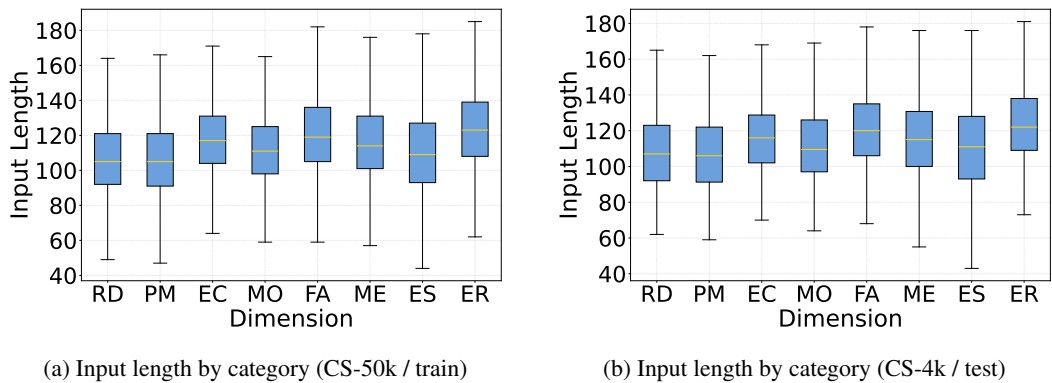

(a) Input length by category (CS-50k / train)          (b) Input length by category (CS-4k / test)

Figure 8: Input length distributions for the training and test splits.

# C TRAINING HYPERPARAMETERS

**SFT** We employ the **LLaMA-Factory** framework (Zheng et al., 2024) for SFT, training the model for three epochs and selecting the checkpoint with the lowest validation loss. A cosine learning rate

schedule with a 10% warmup ratio is applied throughout training. Key hyperparameters, including learning rate, batch size, and weight decay, are summarized in Table 4.

| Hyperparameter | Value |
| --- | --- |
| Learning rate (LR) | $1 \times 10^{-5}$ |
| LR scheduler | cosine |
| Warmup ratio | 0.1 |
| Batch size (per device) | 4 |
| Gradient accumulation steps | 8 |
| Epochs | 3 |
| Weight decay | 0.01 |

Table 4: Hyperparameter settings for SFT training.

**GRPO**  We adopt the **VERL** framework (Sheng et al., 2024) for GRPO training. The actor is initialized from the SFT checkpoint, and rollouts are conducted using vLLM with $n = 6$ sampled responses per prompt. KL regularization is incorporated into the loss with a coefficient of $\beta_{KL} = 10^{-3}$. Training is conducted for three epochs on a single node with 8 GPUs.

Table 5: Key hyperparameters for GRPO training using VERL.

| Hyperparameter | Value |
| --- | --- |
| *Data Configuration* | |
| Max prompt length | 1024 |
| Max response length | 4096 |
| *Algorithm Configuration* | |
| Advantage estimator | `grpo` |
| KL coefficient ($\beta_{KL}$) | $1.0 \times 10^{-3}$ |
| Seed | 1 |
| *Worker: Actor Configuration* | |
| Global batch size | 1024 |
| Micro-batch (update, per device) | 8 |
| Mini-batch (per update) | 16 |
| Learning rate (actor) | $1.0 \times 10^{-6}$ |
| Optimizer | Adam |
| LR warmup ratio | 0.0 |
| *Worker: Rollout Configuration* | |
| Sampler backend | vLLM |
| Rollout trajectories ($n$) | 6 |
| Temperature | 1.0 |
| Top-$p$ | 0.7 |
| Tensor model parallel size | 2 |
| *Trainer Configuration* | |
| Total epochs | 3 |
| Nodes | 1 |
| GPUs per node | 8 |
| Validation frequency (epochs) | 5 |
| Save frequency (epochs) | 8 |

