# OpenReview forum: "ResearchGPT: Benchmarking and Training LLMs for End-to-End Computer Science Research Workflows"
_ICLR.cc/2026/Conference — ICLR 2026 Conference Withdrawn Submission_

### Official Review · Reviewer_kCzr · 2025-10-29

**Soundness:** 2
**Presentation:** 3
**Contribution:** 2
**Rating:** 4
**Confidence:** 3

**Summary:**

This paper proposes two new resources targeting the evaluation and training of large language models in computer science research workflows: CS‑4k, a benchmark for assessing model capabilities across eight stages of the research process, and CS‑50k, a large-scale training set. The datasets are built from 14k CC‑licensed papers from top CS conferences using a retrieval‑augmented generation pipeline, combined with multi‑stage quality control (model‑based reasonability checks, multi‑model filtering, and difficulty scoring). The benchmark aims to provide a holistic view of model performance by covering tasks from problem framing to reporting experimental results. The authors also fine‑tune an open‑source 7B model using CS‑50k with supervised learning and reinforcement learning (GRPO), showing notable gains compared to base performance and in some cases surpassing larger proprietary systems.

**Strengths:**

- The paper addresses an important gap in LLM evaluation by attempting to cover the full spectrum of the research workflow rather than isolated skills.
- The dataset construction pipeline is well‑documented, reproducible, and based on real, high‑quality scientific papers with explicit measures to reduce hallucination through RAG and strict prompt constraints.
- The multi‑stage quality control process is comprehensive, combining automatic checks for reasonability, difficulty balancing, and filtering based on multiple strong models, which helps improve data reliability.
- The release of both a benchmark and a large training set is valuable for the community, enabling not only evaluation but also concrete improvements to domain‑specific capabilities.
- Experimental results demonstrate that domain‑aligned training can yield large gains even for relatively small models, offering practical evidence for efficient capacity building without relying solely on scaling model size.

**Weaknesses:**

- Possible overstatement of “end‑to‑end” evaluation.
  The paper states that CS‑4k supports *end‑to‑end* evaluation of scientific research workflows. In reality, the benchmark combines results from separate sub‑tasks that match different workflow stages. These are tested independently without linked task sequences or shared context, so the process is not a full end‑to‑end pipeline. The claim may therefore be overstated, as the benchmark measures broad coverage rather than full workflow execution.

- Missing variance or uncertainty measures in results.
  The reported CS‑4k scores are given as averages across models, with no measures of variability such as standard deviation or confidence intervals. Without these, it is hard to judge the reliability of the comparisons or to know if the performance gaps between models are statistically significant.

- No check for possible data contamination.
  The paper does not examine whether the CS‑4k set includes material that models may have seen during pre‑training. Without this check, it is unclear if the benchmark tests true generalization to new research data or reflects memorization of known content. This limits confidence in the validity of the comparisons.

**Questions:**

- The paper describes a multi-stage automated quality control pipeline (RAG-based answer grounding, LLM-based reasonability checks, multi-model filtering, and difficulty scoring) to ensure correctness of the QA pairs. However, there is no mention of any human validation or manual sampling to verify the outputs. Could the authors clarify whether any human annotation was performed, and if not, how confident they are that the automated checks are sufficient to catch subtle factual errors or semantic drifts in the generated QA pairs?

- The paper does not provide the exact prompt or input setting used when evaluating model performance on CS‑4k. In particular, it is unclear whether evaluated models were given any retrieved context (e.g., relevant paper passages) during inference, or whether they answered purely from their internal knowledge. Since the presence or absence of context drastically changes the nature of the task and the interpretation of scores, could the authors clarify the evaluation prompt design and whether contextual information was included?

- How is the dataset divided into training and test sets? Is there a possibility that the same paper appears in both, or is there a clear rule ensuring no paper is included in both sets?

---

### Official Review · Reviewer_Tbhp · 2025-10-30

**Soundness:** 2
**Presentation:** 3
**Contribution:** 2
**Rating:** 2
**Confidence:** 4

**Summary:**

Summary: The authors propose CS-4k / CS-50k, a synthetically-generated (through RAG) dataset and benchmark of QA pairs grounded in existing published ML papers. They claim it is the first benchmark to evaluate a model’s ability to assist end-to-end research workflows, a paradigm which they call ResearchGPT. The authors show that frontier models exhibit clear differences in performance on the benchmark based on capability tiers, and also show that training a model on their train set improves performance on their benchmark beyond existing frontier models.

**Strengths:**

1. The authors are tackling an important problem, that being AI-assisted research beyond exam-style questions like in Humanity’s Last Exam (HLE).
2. The authors extract QA pairs from a large corpus of research papers, which is useful as a set of tasks and data for frontier models.
3. The results show clear differences in performance between top frontier models and weaker models on their benchmarks.

**Weaknesses:**

1. The authors claim this dataset / benchmark “systematically evaluates the end-to-end re- search workflow in computer science through open-ended scientific question answering”, but do not justify this claim. The dataset / benchmark consists of synthetically generated QA questions from existing papers, and it is unclear how this connects to the claim.
2. There is a lack of analysis on where the LMs fail / succeed on the tasks, and how to improve existing LM capabilities on these tasks other than training on these specific types of questions.
3. It is not clear other than through LM-as-a-judge filtering why these QA questions are difficult for frontier models. It is also unclear what solving these QA questions would imply.
4. On writing, more examples and emphasis should be placed on the types of questions and answers that the LM generates. From the paper alone, the quantitative metrics do not indicate a lot about the benchmark. For example, it is unsurprising that SFT’ing on the data which is drawn from the same distribution as the train set would increase performance.
5. I would like to see more downstream results of how training on this data affects performance on other related, research-level question benchmarks (e.g. HLE, UQ, etc.).

**Questions:**

1. What was the criteria for splitting the dataset and benchmark? Is it just that the distribution of categories have to match? i.e., can two questions from the same paper be placed into different splits?
2. There is a discussion mini section on “CS-50k training with SFT and GRPO significantly improves model research knowledge.” Are CS-50k and CS-4k completely disjoint with no information leakage?
3. Can you provide concrete examples of questions / answers where models fail? Do models fail because of a lack of knowledge or because of reasoning failures?
4. Can you provide more information on the experiment setup? Is it just a query with no context, or do you also provide some relevant information for the model to use to answer the query?

---

### Official Review · Reviewer_3N7C · 2025-11-01

**Soundness:** 2
**Presentation:** 3
**Contribution:** 2
**Rating:** 4
**Confidence:** 3

**Summary:**

The paper introduces ResearchGPT, a framework and benchmark designed to evaluate and train language models as scientific research assistants. The authors construct CS-54k, a large corpus of research-grounded Q&A derived from ~14k computer science papers using a retrieval-guided, multi-stage quality-control pipeline. From this, they derive CS-4k, a curated evaluation benchmark aligned with real research workflows, and CS-50k, a training set for model alignment.

Using these resources, the authors fine-tune Qwen2.5-7B-Instruct, showing substantial gains over its base model and competitive performance against several larger proprietary LLMs. They also demonstrate that adding reinforcement learning with dual reward models further improves reasoning quality and reduces reward hacking. The work argues that domain-aligned data and alignment strategy matter more than scaling alone for building research-capable LLMs.

**Strengths:**

- This work contributes lots of QA pairs for computer science papers
- The proposed trained 7B model outperforms several proprietary LLMs that are much larger than 7B
- The proposed data construction pipeline may benefit the research community
- The benchmark reveals some interesting findings on LLMs' capabilities in answering questions related to computer science papers

**Weaknesses:**

- The question-answering setting seems too naive. It seems that the questions in the benchmark don't contain the raw text from related papers (Figure 5). I don't think this is a very useful setting. It would be more meaningful if you append the entire paper or append the same text chunk with the question.
- The comparison is not very fair. The answer is generated by LLM based on guidelines, which introduces a bias. The guidelines for generating the answer at test time is not introduced in the question, I worry this benchmark leans towards testing how the output answer is aligned with the guideline and LLM judge's preference, instead of LLM's real ability.
- The wording is a bit misleading. Many phrases in this paper give an impression that LLM is conducting research by itself, for instance, the title and line 362 "CS-4k cleanly separates capability tiers of SOTA LLMs in scientific research".

**Questions:**

- At test time, do you provide models with access to the full paper (or retrieved text chunks) via retrieval augmentation, or are the evaluations performed without any context?
- In line 362, is "cleanly" a typo?

---

### Official Review · Reviewer_uAAo · 2025-11-03

**Soundness:** 1
**Presentation:** 3
**Contribution:** 2
**Rating:** 2
**Confidence:** 3

**Summary:**

The paper builds a Q&A dataset (e.g., "What's the main point of paper XYZ" or "What hardware was used when evaluating models for the XYZ project") out of open source licensed CS papers. Models are evaluated based on whether they can answer this question directly.

## Dataset construction

**Question gen**

1. basic questions are sampled based on topic classes
2. the question is augemented/different versions are created
3. the finally question is created based on the augmented question together with context obtained by RAG.

**Answer gen**: Answer is obtained using a prompt based on RAG over the crawled papers


**Validation**: The question answer pairs (Q, A) are validated by

1. LM as a judge using deepseek R1, whether A fits to Q
2. Take `(Q, A)` pairs. Let GPT 4 mini, gemini 2.5 flash, and Claude 3.5 haiku answer the question `Q` as `A'`. Then we prompt another LM (I assume R1?) to give scores based on `(Q, A, A')`. If all models are correct, the `(Q, A)` pair is removed. If all models are incorrect, `(Q, A)` is also dropped. This is meant to reserve trivial and faulty task instances.

## Experiments

From this pipeline, two subsets are constructed: CS-4k (serving as benchmark) and CS-50k (serving as training dataset)

The authors then perform training experiments, finetuning & RL training LMs on CS-50k, then observing score improvements on CS-4k.

**Strengths:**

The paper constructs a large corpus of Q & A pairs for computer science literature. While mostly not a completely new concept, this corpus might be useful training data. A subset of this corpus is used as a benchmark for Q&A on computer science questions and might be useful when evaluating models on their CS paper knowledge, though the evaluation should be validated thoroughly.

The authors also experiment with finetuning and RL and show that the training corpus can lead to higher scores on the benchmark.

**Weaknesses:**

* The filtration step where `(Q, A)` pairs are removed if GPT 4 mini, Gemini 2.5 flash or Claude 3.5 haiku either are all correctly or are all incorrect (based on LM as a judge given the ground truth answer) requires justification. Does this not introduce significant bias into the CS-4k benchmark? For the training dataset, this probably doesn't matter, but for the benchmark CS-4k it might especially be problematic that comparatively weak models (compared to e.g., o3 etc.) are used for this filtration. There are also no explicit on how many problems are removed in this step, but it seems like around half? I'm not sure this filtration step is beneficial for CS-4k. I don't think there's any problem with having easy instances in the benchmark. And I don't think this step can distinguish well between "too hard" vs "faulty" instance. It might also still be better to have some faulty instances compared to much more complicated biases.
* This benchmark relies heavily on LM (I assume deepseek r1?) as a judge but there is not much information on what was done to verify that it is behaving as expected. For example by having humans independently score a sample and comparing scores, or manually investigating a small set of classification results (and reasoning thereof), by looking at the variance when repeatedly evaluating the same answer, by looking whether rephrasing the same answer can change the score etc. Looking at the scoring template I also wonder how things like "LMs gives the main point but adds other information that's not corroborated by the sources" is handled (in other words, is it a winning strategy to just dump as much information as possible). Basically there's both a  precision and a  recall component and it's not clear in the prompt how that should be converted into a single score.
* The training experiments are only performed on the CS-4k benchmark, which is exactly constructed like the CS-50k training data. This could be very biased, especially because of the previous two points. Are there any other completely independent benchmarks that could be evaluated on to demonstrate that there is a true improvement in model quality and not just some sort of fitting to the biases of the benchmark?

**Questions:**

* In Fig. 5 could you add the scores that the models get? Histograms of the scoring would also be helpful (i.e., how often does the LM judge put answers into the highest category etc.). In general, any supporting data that shows validation of the LM as a judge and the filtration process would be helpful.
* Which models are used in dataset constructions and in the LM as a judge setup? It seems like this is probably DeepSeek R1, but for example the model is not explicitly mentioned in line 239, 245 etc., or most importantly for the evaluation prompt. Did you check consistency when swapping out R1 with other models?
* Since the template "prompt for evaluation" returns numbers, what are the exact decision boundaries for the filtration in line 239+ (where Q, A pairs are excluded if sample models either are all correct or all incorrect)?
* Would it be possible show any evidence of improvement of the finetuned/RL trained model on other benchmarks?

---

### Note · Authors · 2025-11-13

I have read and agree with the venue's withdrawal policy on behalf of myself and my co-authors.